# Systematical Investigation of Flicker Noise in 14 nm FinFET Devices towards Stochastic Computing Application

**DOI:** 10.3390/mi14112098

**Published:** 2023-11-14

**Authors:** Danian Dong, Jinru Lai, Yan Yang, Tiancheng Gong, Xu Zheng, Wenxuan Sun, Jie Yu, Shaoyang Fan, Xiaoxin Xu

**Affiliations:** 1State Key Laboratory of Fabrication Technologies for Integrated Circuits, Institute of Microelectronics of the Chinese Academy of Sciences, Beijing 100029, China; 2School of Microelectronics, University of Chinese Academy of Sciences (UCAS), Beijing 101408, China; 3School of Microelectronics, University of Science and Technology of China, Hefei 230026, China; 4College of Communication Engineering (College of Microelectronics), Chengdu University of Information Technology, Chengdu 610225, China

**Keywords:** stochastic computing, 14 nm FinFET, flicker noise, SNG

## Abstract

Stochastic computing (SC) is widely known for its high error tolerance and efficient computing ability of complex functions with remarkably simple logic gates. The noise of electronic devices is widely used to be the entropy source due to its randomness. Compared with thermal noise and random telegraph noise (RTN), flicker noise is favored by researchers because of its high noise density. Meanwhile, unlike using RRAM, PCRAM and other emerging memory devices as the entropy source, using logic devices does not require any additional process steps, which is significant for industrialization. In this work, we systematically and statistically studied the *1/f* noise characteristics of 14 nm FinFET, and found that miniaturizing the channel area of the device or lowering the ambient temperature can effectively increase the *1/f* noise density of the device. This is of great importance to improve the accuracy of the SC system and simplify the complexity of the stochastic number generator (SNG) circuit. At the same time, these rules of *1/f* noise characteristics in FinFET devices can provide good guidance for our device selection in circuit design.

## 1. Introduction

As device scaling continues to nanoscale dimensions, the reliability of integrated circuits will continue to become a greater problem. The expected higher probability of failure as well as the higher sensitivities to variations could make future integrated circuits more unreliable [1,2,3]. Stochastic computing (SC), an operation based on stochastic bit streams (SBS), is broadly similar to brain’s neural spike sequences. Its key feature is the use of low-cost and low-power logic elements to implement complex numerical operations in a highly fault-tolerant manner [4,5,6]. Moreover, SC can be implemented for several useful applications effectively such as image processing [7], scientific computing [8] and neural network computing, etc. In 2020, for example, professor Kaushik Roy [9] discussed how to use the stochastic sigmoid-like characteristics of spin-orbit torque magnetic tunnel junctions (SOT-MTJ) to realize stochastic neural networks. Finally, they observed that in a network with full-precision weights, having stochastic binary activations reduces the energy consumption by ~40%. In addition, networks with binary weights achieve ~28× improvement in energy consumption compared with a typical CMOS-based von Neumann system. Nevertheless, the accuracy of SC severely depends on the length and correlation of SBS [10]. Therefore, it is highly demanded to find a stochastic number generator (SNG) to generate long and independent SBS, which is critical for the accuracy of the SC system.

At present, there are two main ways to implement SNG: one is the linear feedback shift registers (LFSR)-based pseudo stochastic number generator (PSNG) [11,12,13,14], as shown in Figure 1a. It can generate a pseudo-random sequence with a period of 2n − 1 by using an LFSR circuit composed of n flip-flops. The hardware implementation of this method is very convenient. However, it generates periodic pseudo-random numbers, which will have a bad impact on the calculation accuracy of SC. In 2020, for example, professor Salehi [14] proposed an area-efficient SNG by sharing the permuted output of one linear feedback shift register among several SNGS. Although this method reduces the circuit area, it requires increased hardware complexity to reduce the correlation between the generated bit streams. The other one is to use internal noise of electronic devices (such as MOS devices or emerging memory devices) to realize SNG as shown in Figure 1b. In 2021, for example, our team [15] proposed a 128 kb RRAM flicker-noise-based stochastic computing chip. The basic idea is to use flicker noise in RRAM as the entropy source for SNG, and finally this RRAM-flicker-noise-based SC chip is successfully implemented for image edge detection. Compared with emerging memory devices, MOS devices do not need additional processes, which is more convenient and economical. Generally, there are mainly three kinds of noise including thermal noise, RTN and flicker noise. All of these three kinds of noise have been exploited as entropy source due to its natural randomness, as shown in Figure 2a,b. Compared with thermal noise [16], RTN [17,18], flicker noise [19] has the advantage of high noise density, which is beneficial for the SC application. Thus, it is important to study SNG based on the flicker noise of MOS devices for the industrialization of the SC system.

Up to now, most articles focus on how to reduce the flicker noise [19,20,21]. However, for the SC application, it is essential to acquire noise with high density. In this work, we systematically and statistically study the flicker noise characteristics of 14 nm FinFET devices. It is found that the channel area of logic devices and the operating temperature are the two key factors to improve the noise density of the MOS devices.

## 2. Device Fabrication and Experiments

The FinFET devices are fabricated based on the Semiconductor Manufacturing International Corporation (SMIC) 14 nm FinFET CMOS process platform. The Fin width (*T_fin_*) and height (*H_fin_*) were 12 nm and 21 nm, respectively, and the effective device width was approximately given by *N_finger_* × *N_fin_* × (2 × *H_fin_* + *T_fin_*). So, the channel area of the FinFET device is calculated as follows:(1)$$A=W_{eff}\times L=N_{fin}\times N_{finger}\times \left(2\times H_{fin}+T_{fin}\right)\times L$$

Here, *A* is the channel area, *W_eff_* is the effective device width, and *L* is the channel length

A series of experiments was performed to characterize the electrical parameters of the tested devices, including the *I_D_*-*V_G_* tests and the *1/f* noise tests. Before measuring *1/f* noise, we first need to measure the *V_th_* value of the device under test. At this time, the device under the test is connected to the Agilent B1500 semiconductor parameter analyzer through the SUMMIT 12000B probe station. Then, scan the *I_D_*-*V_G_* curve of the device and calculate the *V_th_* value of the device through the extrapolated threshold voltage method. Next, we need to disconnect the device from the B1500, then connect it to the Proplus 9821DX noise measurement system through the probe station, apply bias to the gate and drain of the device, *V_G_* = *V_th_*, *V_D_* = 0.1 V, and then start measuring *1/f* noise.

## 3. Results and Discussion

In the characteristics of *1/f* noise in 2D logic devices, researchers generally believe that the normalized drain current noise (*A* × *S_ID_*/*I_D_*^2^) is approximately a constant when *V_GS_* equals to *V_th_*, and the amplitude of normalized drain current noise (*S_ID_*/*I_D_*^2^) is inversely proportional to the area of the device channel [19]. Here, *A* is the area of the device channel, *S_ID_* is the current noise of the drain end, and *I_D_* is the current of the drain end. In order to systematically explore the *1/f* noise characteristics of 3D logic devices, we have made a statistical study of three different FinFET devices, namely 4Fin-2Finger, 4Fin-1Finger and 2Fin-1Finger. As shown in Figure 3a, the normalized drain current noise (*A* × *S_ID_*/*I_D_*^2^) is also approximately a constant in three FinFET devices with different structures. Compared with the other three 2D logic devices, namely HKMG with a 28 nm process, PolySiON with a 28 nm process and HLMC with a 40 nm process, it is found that the normalized drain current noise (*A* × *S_ID_*/*I_D_*^2^) parameters of devices with different process nodes, different materials and different structures are all approximately a constant, as shown in Figure 3b. This shows that the amplitude of the normalized drain current noise (*S_ID_*/*I_D_*^2^) has nothing to do with the process node, material and structure, but only with the area of the channel, whether in 2D or 3D logic devices. Therefore, reducing the channel area is one of the effective ways to obtain high-density noise. In addition, we can see from Figure 3b that the variance of the normalized drain current noise (*A* × *S_ID_*/*I_D_*^2^) of devices with different process nodes is different, and as the process node decreases, the variance of the normalized drain current noise (*A* × *S_ID_*/*I_D_*^2^) also becomes smaller. A possible reason is that as process nodes are scaled down, the variability in defect density distribution in the device channels also decreases.

In order to explore other methods to improve the amplitude of *1/f* noise, we further study the origin of *1/f* noise of 3D FinFET devices. Currently, there are two main physical models to explain *1/f* noise, McWhorter’s model [22] (noise caused by carrier number fluctuation, CNF), described with Formulas (2) and (3), and Hooge’s model [23] (noise caused by mobility fluctuation, MF), described with Formula (4).
(2)$$\;\;\;\;\frac{S_{I_{D}}}{{I_{D}}^{2}}={\left(\frac{g_{m}}{I_{D}}\right)}^{2}\times S_{VFB}$$
(3)$$S_{VFB}=\frac{q^{2}kT\lambda N_{t}}{WL{C_{ox}}^{2}f}$$

Here, *g_m_* is the transconductance, $I_{D}$ is the drain current, and $S_{VFB}$ is the flat band voltage noise. $S_{VFB}$ is as shown in Formula (3), where q is unit charge, k is the Boltzmann constant, T is the temperature, *λ* is the tunneling attenuation length, $N_{t}$ is the trap density, W is the channel width, L is the channel length, $C_{ox}$ is the oxide capacitance, and f is the frequency.
(4)$$\frac{S_{ID}}{{I_{D}}^{2}}=\frac{qa_{H}u_{eff}V_{D}}{fL^{2}I_{D}}\;$$

Here, $a_{H}$ is the Hooge parameter. *V_D_* is the bias voltage on the drain end.

From Formula (2), we know that when carrier number fluctuation (CNF) plays a major role, the trends of normalized drain current noise ($S_{ID}$/${I_{D}}^{2}$) and (*g_m_*/*I_D_*)^2^ are the same. On the other hand, according to Formula (4), normalized drain current noise (*S_ID_*/*I_D_*^2^) is inversely proportional to the drain current. As shown in Figure 4a, we measured the noise density curves under different *V_GS_* biases. Meanwhile, the relationship between noise density and the *I_D_* of the device at 1000 Hz is drawn in Figure 4b. The trend of noise density and *I_D_* is the same as that of its (*g_m_*/*I_D_*)^2^ and *I_D_*. 

Therefore, for 3D FinFET devices, McWhoter’s carrier number fluctuation model plays a major role, which is the same as that of 2D logic devices.

According to McWhoter’s carrier number fluctuation model, temperature and trap density can significantly affect the noise amplitude. In order to study the effect of temperature, we measured the noise density curves of the devices in the temperature range of 218–298 K, as shown in Figure 5a. It can be seen that normalized drain current noise (*S_ID_*/*I_D_*^2^) of 3D FinFET devices is inversely proportional to temperature. Generally speaking, low-frequency noise or *1/f* noise is considered to be the superposition of multiple RTNs of different time constants. RTN and *1/f* noise are known to be caused by defects (or traps) close to the silicon–oxide interface. These defects (or traps) capture and release carriers from and into the inversion channel, causing fluctuations in the drain current. Recent observations show that oxide relaxation is now considered to be the determinant factor for a carrier to be trapped [24,25]. The average time to capture, ${\bar{\tau }}_{c}$, is modeled as
(5)  τ¯c=1nsσv¯th=1nsσ0v¯theEBkT

Here, *n_s_* is the channel carrier density, ${\bar{v}}_{th}$th is mean thermal voltage of carriers, and σ is capture cross-section. $\sigma_{0}$ is a scaling parameter. *k* is the Boltzmann constant and *T* is the absolute temperature. *E_B_* is the energy barrier for a carrier to be trapped. With the increase in temperature, an exponential decrease in ${\bar{\tau }}_{c}$ is expected according to the model. In addition, time to emission, ${\bar{\tau }}_{e}$, is modeled as
(6)τ¯e=τ¯c×e−(ET−EF)kT

Similarly, ${\bar{\tau }}_{e}$ also decreases exponentially with the increase in temperature. Therefore, at the condition of low temperature and normal temperature, as the temperature rises, these defects (or traps) at the interface capture and release electrons faster. In this condition, some of these defects (or traps) cannot be captured. As a result, the density of traps that can contribute to *1/f* noise is reduced, which ultimately leads to a reduction in *1/f* noise.

At the same time, we also measured the *I_D_*-*V_GS_* curve of the device in the temperature range of 218–298 K and calculated the relationship between carrier mobility and temperature as shown in Figure 5b,c. It can see that the mobility decreases with the increase in temperature in the linear region (*V_DS_* = 0.1 V), which is consistent with the trend of noise density and temperature. This result further verifies that the 3D FinFET devices conforms to McWhoter’s model. Thus, reducing the ambient temperature provides a new way to obtain larger noise without reducing the channel area of the device.

## 4. Conclusions

In this work, we systematically and statistically studied the *1/f* noise characteristics of 3D FinFET devices. First of all, we find that the normalized drain current noise (*A* × *S_ID_*/*I_D_*^2^) is approximately a constant for both 2D and 3D logic devices, which indicates that the amplitude of the normalized drain current noise (*S_ID_*/*I_D_*^2^) has nothing to do with the process node, material and structure of the device, but only with the channel area of the device. Secondly, the low-temperature environment can effectively increase the noise amplitude of the device. These methods can serve as a good guide, ultimately allowing the *1/f* noise of MOSFET devices to be better applied in stochastic computing systems. For example, on the one hand, when we design the SNG circuit of the SC system, we can choose devices with as few fingers and fins as noise source devices. In this way, the smaller the channel area of the device, the greater the amplitude of its intrinsic *1/f* noise. A larger noise amplitude is beneficial to the parameter design of the subsequent stage noise sampling circuit and the reliability of the circuit. On the other hand, when we design other peripheral circuits of the SC system, we can choose as many fingers and fins devices as possible, so that the intrinsic *1/f* noise of the device can be well suppressed by increasing the device channel area, thus further improving the reliability of the overall circuit.

## Figures and Tables

**Figure 1 micromachines-14-02098-f001:**
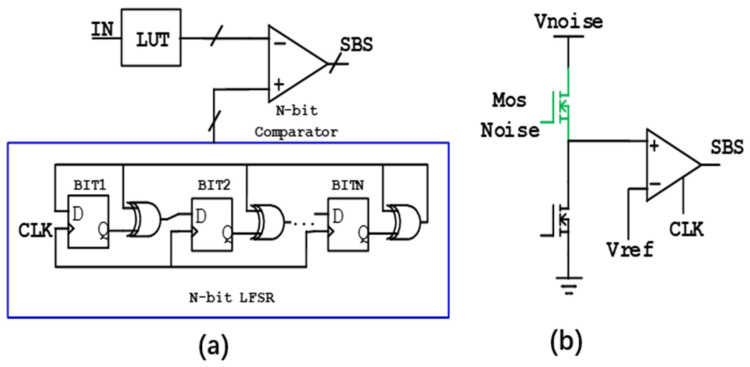
(**a**) LFSR-based PSNG. (**b**) MOS noise-based SNG.

**Figure 2 micromachines-14-02098-f002:**
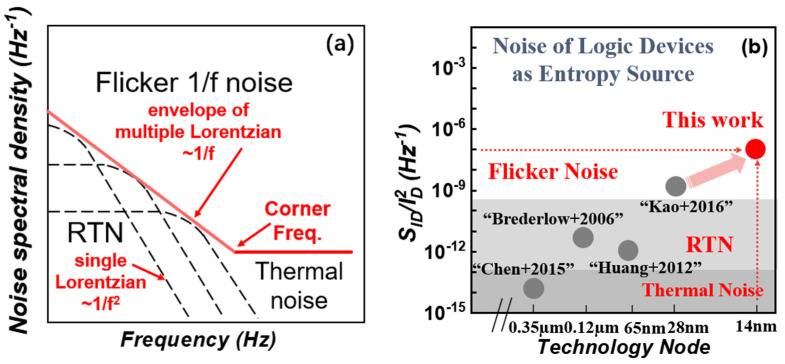
(**a**) Illustration of different noise types in electronic devices. (**b**) Comparison of different noise types as entropy sources [16,17,18,19].

**Figure 3 micromachines-14-02098-f003:**
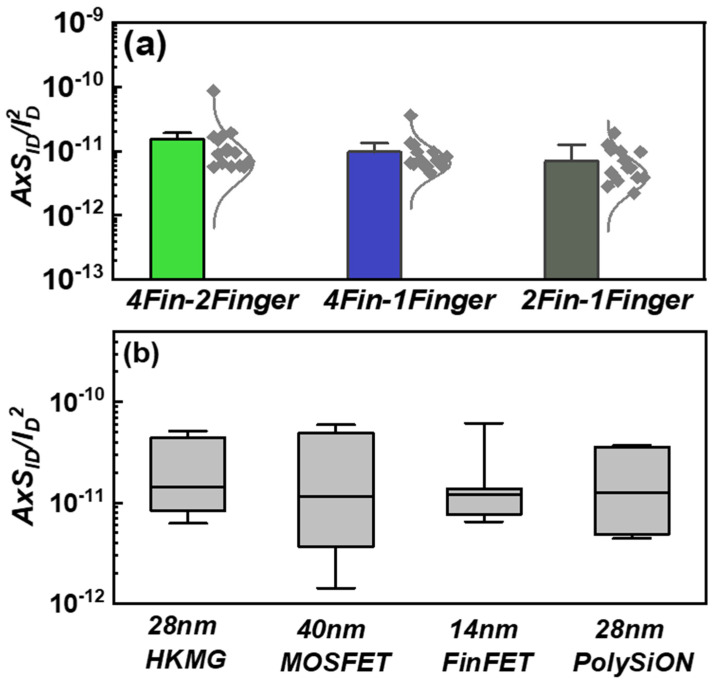
(**a**) Normalized noise level of three FinFET devices with different structures at *f* = 1000 Hz, which are 4Fin-2Finger, 4Fin-1Finger and 2Fin-1Finger. (**b**) Normalized noise level of four different devices at *f* = 1000 Hz, which are 28 nm HKMG MOSFET, 40 nm MOSFET, 14 nm FinFET, and 28 nm PloySiON MOSFET.

**Figure 4 micromachines-14-02098-f004:**
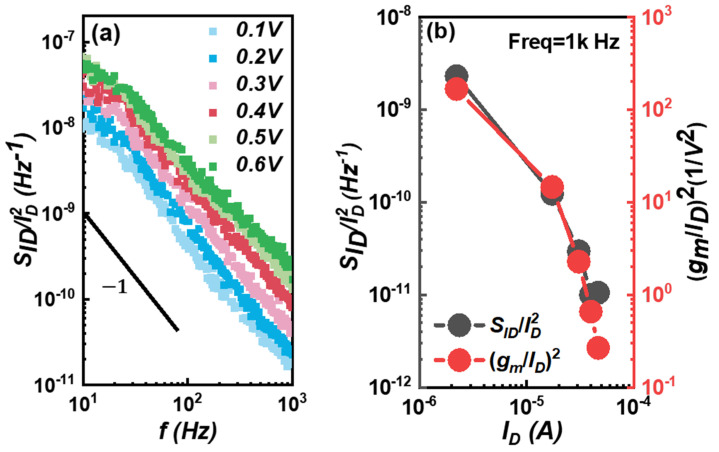
(**a**) The *1/f* noise results of the FinFET device in the subthreshold region; the *V_G_* voltage ranges from 0.1 V to 0.6 V in step of 0.1 V. (**b**) Normalized drain current noise (*S_ID_*/*I_D_*^2^) has the same trend as (*g_m_*/*I_D_*)^2^, which indicates the *1/f* noise behaviors can be interpreted with McWhoter’s model.

**Figure 5 micromachines-14-02098-f005:**
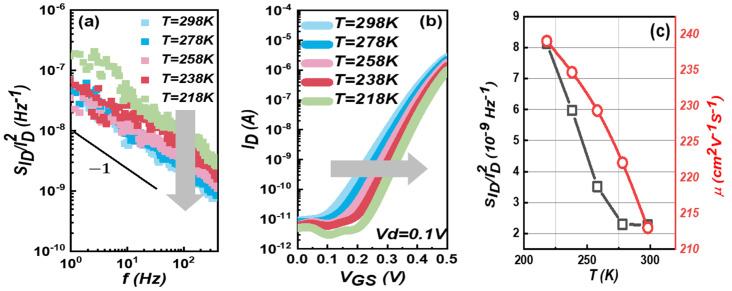
(**a**) The *1/f* noise curves of FinFET device under different temperatures in the range from 218 K to 298 K. Both show the temperature dependence. (**b**) The *I_D_*-*V_GS_* transfer curves. (**c**) *T* dependence of field-effect mobility *µ* (red) and the normalized power spectral density *S_ID_*/*I_D_*^2^ (black).

## Data Availability

The data that support the findings of this study are available from the corresponding author upon request.

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
