# Peer review of "Systematical Investigation of Flicker Noise in 14 nm FinFET Devices towards Stochastic Computing Application"

_micromachines, 2023, doi:10.3390/mi14112098_

Round 1

Reviewer 1 Report

Comments and Suggestions for Authors

Question 1: In Fig. 3(a), it would be good to elaborate on why the results of comparing the number of fins (Nfin) and fingers (Nfinger) are constant. It would be helpful to elaborate on why the three cases are similar regarding what characteristics lead to less variability.

Question 2: It is said that 1/f noise is constant depending on the material and structure and device tech node (Fig. 3 (b)), but 1/f changes depending on mobility, channel charge variation, SiO2-Si interface quality, and the impact of 1/f increases as the device size decreases, so I doubt that the 1/f noise described in the text is constant. A more qualitative comparison is needed. In addition, as the technology node of the device decreases, the bias also scales down, and the noise characteristics change due to the change in current under voltage conditions, so it doesn't seem easy to make an accurate comparison.

Question 3: If the 14 nm node FinFETs are more resistant to flicker noise, it would be nice to have something to compare to the 14 nm node to give more confidence in the results in Fig. 4.

Question 4: It would be good to add a more detailed explanation of Fig. 4 (b), and if the major role of the carrier number fluctuation model of 2D logic devices is followed, does it mean that there is no improvement without the difference in flicker noise between 2D logic devices and FinFET logic devices?

Question 5: In Fig. 5(a), it would be helpful to include the temperature-dependent 1/f variation data of the 2D device to compare it to the FinFET and explain how it differs.

Question 6: Add an explanation for why interface trap density changes with temperature.

Question 7: The conclusion states that the 1/f noise is constant for 2D and 3D devices. So, is the main point of this paper to maintain the noise characteristics while increasing the integration rate due to scaling down the device? The purpose and justification for going to FinFET devices need to be improved. It would be better to supplement it with a higher-quality paper.

Author Response

Dear Reviewer:

         Thank you very much for your valuable comments. After careful consideration, we have written the point-to-point response in the attached document. please see the attachment.

Thank you!

Reviewer 2 Report

Comments and Suggestions for Authors

This paper investigates 1/f noise for advanced FinFET device. It is a very excellent paper and can be accepted in its current form. Authors may or may not optionally add the information below.

1) Recently, there has been a lot of research on the application of neuromorphic systems using stochastic switching, including by Professor Roy's team at Purdue University. I hope the introduction will be strengthened in this regard.

2)The link between 1/f noise and stochastic switching is not clear. Strengthen this part

3) There is a need to discuss what vulnerabilities the FinFET structure has to noise compared to the existing planar structure.

4) It would also be good to have some discussion about how scalability is possible in structures such as nanosheets and MBCFETs.

Author Response

Dear Reviewer:

         Thank you very much for your valuable comments. After careful consideration, we have written the point-to-point response in the attached document. Please see the attachment.

Thank you!

Round 2

Reviewer 1 Report

Comments and Suggestions for Authors

My Q&A is fully satisfied.